# Scalable Bayesian Optimization via Focalized Sparse Gaussian Processes

**Yunyue Wei[1], Vincent Zhuang[2], Saraswati Soedarmadji[1], Yanan Sui[1]**
[1] Tsinghua University
[2] Google DeepMind
weiyy20@mails.tsinghua.edu.cn
vincentzhuang@google.com
chenxuying24@mails.tsinghua.edu.cn
ysui@tsinghua.edu.cn

## Abstract

Bayesian optimization is an effective technique for black-box optimization, but its applicability is typically limited to low-dimensional and small-budget problems due to the cubic complexity of computing the Gaussian process (GP) surrogate. While various approximate GP models have been employed to scale Bayesian optimization to larger sample sizes, most suffer from overly-smooth estimation and focus primarily on problems that allow for large online samples. In this work, we argue that Bayesian optimization algorithms with sparse GPs can more efficiently allocate their representational power to relevant regions of the search space. To achieve this, we propose focalized GP, which leverages a novel variational loss function to achieve stronger local prediction, as well as FocalBO, which hierarchically optimizes the focalized GP acquisition function over progressively smaller search spaces. Experimental results demonstrate that FocalBO can efficiently leverage large amounts of offline and online data to achieve state-of-the-art performance on robot morphology design and to control a 585-dimensional musculoskeletal system.

## 1 Introduction

Bayesian Optimization (BO) is a powerful approach for solving black-box optimization problems, demonstrating notable success in hyperparameter tuning[1], reinforcement learning[2, 3], and scientific discovery[4]. The efficacy of BO is attributed to its ability to model the unknown objective function using a surrogate model and to strategically select the next sample position by optimizing an acquisition function. Among the surrogate models, Gaussian Processes (GPs)[5] are usually favored due to their flexibility and robust uncertainty quantification. However, the computation of the posterior GP covariance matrix scales as $\mathcal{O}(n^3)$ with the number of data points $n$, which can severely restrict the applicability of BO in handling large datasets. This poses a significant challenge for real-world applications with high-dimensional and heterogeneous function landscapes such as those in robot control, which often necessitate a substantial amount of data to adequately explore the vast search space. To extend the scope of BO to accommodate larger datasets (from long-horizon online trials and/or pre-collected offline datasets[6]), it is imperative to employ surrogate models that offer enhanced computational efficiency.

Using sparse GP models is a popular method for reducing the computational cost of BO. Sparse GPs accomplish this by learning an approximation of the full GP, either by using a

38th Conference on Neural Information Processing Systems (NeurIPS 2024).

subset of data[7], ensemble of local models[8], or variational inference[9]. However, classical sparse GP models are typically tailored for regression tasks, and therefore are designed to fit to the entire function landscape. Given limited representational resources, the resulting posterior is likely to be overly smooth, which may negatively impact the performance of BO. This issue is exacerbated in the high-dimensional setting, in which accurately fitting the entire domain is a far more challenging task. As such, several works have proposed strategies to improve BO performance with sparse GP models by focusing promising regions[10, 11] or advanced sparse GP models[12]. However, most of their empirical evaluations are only conducted under large online sample setting in low-dimensional problems with fewer than 20 variables. It is unclear whether existing methods can be generalized to large offline data or high-dimensional setting.

In this work, we explore the application of sparse Gaussian processes for optimizing high-dimensional problems with large offline (and optionally large online) datasets. We argue that by iteratively identifying key sub-regions of the input space and focusing the modeling capacity on these areas, we can enhance the modeling fidelity of the sparse GP in regions that are most relevant, thereby improving the overall performance of the Bayesian optimization algorithm. To this end, we propose a novel loss function to train a variational sparse GP model (focalized GP) that emphasizes the fitting of local functional landscapes through weighting the training data. Along with focalized GP, we design a hierachical algorithm, FocalBO, to propose sample points via acquisition function optimization across varying scales of the search space. Experimental results demonstrate that FocalBO can improve upon commonly used acquisition functions in optimizing heterogeneous functions and can effectively utilize large offline datasets for efficient high-dimensional optimization. Furthermore, we showcase that FocalBO can efficiently optimize a policy with 585 parameters to control a musculoskeletal system, leveraging both offline and online data. To the best of our knowledge, FocalBO is the first sparse GP-based Bayesian optimization algorithm capable of efficiently optimizing high-dimensional problems under both large online sample and large offline data settings.

Our main contributions: 1) We design FocalBO, which employs a hierarchical acquisition optimization strategy to achieve efficient optimization over high-dimensional problems with heterogeneous structure with limited representation capability. 2) Experimental results demonstrate the superior performance of FocalBO in leveraging large offline datasets for online optimization, and its capability to optimize high-dimensional musculoskeletal system control problems involving over 500 variables.

## 2   Related Work

### 2.1   Sparse Gaussian processes

Scaling Gaussian processes to large datasets is an important topic [13]. It can be broadly divided into global approximation strategies and local approximation strategies.

Global sparse GPs perform distillation over the whole dataset to approximate the expensive full covariance matrix with a sparse representation. Several methods aim to choose a subset of representative training points from the whole dataset, and use the corresponding covariance matrix in place of the full covariance[14, 7, 15, 16]. Sparse kernels aim at removing uncorrelated entries in the full covariance to obtain a compact matrix[17, 18, 19, 20]. Sparse approximation methods use inducing variables to learn a low-rank representation of full covariance matrix[21, 22, 23, 24, 8, 9, 25, 26, 27]. Stochastic variational Gaussian process (SVGP) is a popular sparse GP method which employs variational inference to learn inducing variables and kernel hyperparameters jointly and enable training using stochastic gradient descent from mini-batch data[25]. Recently, nearest neighbor information has also been used to further improve the scalability of sparse GP over massive amount of data[28, 29].

In contrast, local sparse GPs divide the entire dataset and employ local GPs trained from different data subsets to approximate the full GP. For a given test set, the prediction can be extracted from one of the local GPs[30, 31], mixture of GPs[32, 33] or product of GPs[34, 35].

## 2.2 Scalable Bayesian optimization

Recent works have proposed modifications to sparse GPs for Bayesian optimization. Sparse GP has been used to determine the search region where local GPs are used to determine the next samples[36]. Weighted-update online Gaussian processes (WOGP) was developed to select a subset of training points to approximate high performing regions of the input space[10]. IMP-DPP is motivated by a similar observation and uses a weighted Determinantal Point Process to select training points as inducing variables for the SVGP[11]. However, their proposed selection strategies require sequentially evaluating every training point, which can be computationally very expensive with large offline datasets. Combining SVGP with Thompson sampling has the same order of regret as standard Thompson sampling method[37]. Online variational conditioning (OVC) was proposed to efficiently conditioning SVGPs in an online setting, enabling using look-ahead acquisition functions[38]. Vecchia approximation of GP was also applied[39] for Bayesian optimization, with improved performance compared to prior works[12]. A concurrent work [40] aims at improving the acquisition optimization performance based on target-aware Bayesian inference [41].

Besides sparse GPs, Neural network[42, 43] and random forest[44] can also be used as BO surrogate model to circumvent the cubic complexity of GP. Ensemble Bayesian optimization utilizes the addictive function structure and uses ensembles of addictive GPs in parallel to achieve scalability[45]. Trust Region Bayesian optimization (TuRBO) and its variants uses exact GP to optimize over local regions, and employs a restart mechanism to achieve large number of evaluation, which is a representative line of works in high-dimensional Bayesian optimization[46, 47, 48]. TuRBO can also be combined with sparse GP models to further enhance the scalability[49, 50].

## 3 Background

### 3.1 Bayesian optimization

For an unknown objective function $f$, Bayesian optimization aims to solve $\max_{\boldsymbol{x} \in \mathcal{X}} f(\boldsymbol{x})$ over input space $\mathcal{X} \in [0, 1]^d$. BO mainly consists of two components: a surrogate model to approximate the objective function, and an acquisition function $a$ to decide the next sample position based on surrogate model.

Gaussian process is a commonly used surrogate model. Consider a given dataset $D = (\boldsymbol{X}, \boldsymbol{y})$ where $\boldsymbol{X} = (\boldsymbol{x}_1, ..., \boldsymbol{x}_t)$ are input locations and $\boldsymbol{y} = (y_1, ..., y_t)$ are associated noisy observations of $f(\boldsymbol{X})$. We assume the observation noise to be independent Gaussian, i.e. $y_i = f(\boldsymbol{x}_i) + \eta, \eta \sim \mathcal{N}(0, \sigma^2)$. Using GP with kernel function $K$, the function distribution $\boldsymbol{f}_*$ at test positions $\boldsymbol{X}_* = (\boldsymbol{x}_{*,1}, \ldots, \boldsymbol{x}_{*,t_*})^T$ is a multivariate Gaussian:

$$p(\boldsymbol{f}_* \mid \boldsymbol{X}, \boldsymbol{y}) = \mathcal{N}(\boldsymbol{f}_* \mid K_{\boldsymbol{X}_* \boldsymbol{X}}[K_{\boldsymbol{X} \boldsymbol{X}} + \sigma I]^{-1}\boldsymbol{y},$$
$$K_{\boldsymbol{X}_* \boldsymbol{X}_*} - K_{\boldsymbol{X}_* \boldsymbol{X}}[K_{\boldsymbol{X} \boldsymbol{X}} + \sigma I]^{-1}K_{\boldsymbol{X} \boldsymbol{X}_*}), \tag{1}$$

where $K$ is the covariance matrix between subscript inputs. With the posterior distribution given $D$, the next sample point is the maximum position of the acquisition function: $\boldsymbol{x}_{t+1} = \max_{\boldsymbol{x} \in \mathcal{X}} a(\boldsymbol{x}|\mathcal{M}_t)$, where $\mathcal{M}_t$ is the GP model fitted on dataset collected at time step $t$. Common-used choice of $a$ includes upper confidence bound(UCB, [51]), expected improvement (EI, [52]) and Thompson sampling (TS, [53]). The inner optimization problem is usually solved by grid search, evolutionary algorithms[54], or gradient-based methods[55]. When the online sample budget is large, batch optimization is commonly used to evaluate multiple inputs in parallel[56].

### 3.2 Variational Gaussian process

For predictive distribution conditioned on given dataset of size $t$, the computational complexity of exact Gaussian process is $\mathcal{O}(t^3)$ for each test position due to the inverse of the covariance matrix $K_{\boldsymbol{X} \boldsymbol{X}}$, which is expensive for large scale datasets with more than a few thousand points. A common used strategy is to approximate full GP regression using sparse GPs. In sparse GP, $m \ll t$ inducing variables $\boldsymbol{u} = (u_1, \ldots, u_m)^T$ characterized by inducing inputs

$Z = (z_1, \ldots, z_m)$ are introduced to approximate the covariance matrix of the full GP. In this section, we focus on sparse GP derived from variational inference.

Variational GP [9] considers the joint latent prior

$$p(\boldsymbol{f}, \boldsymbol{u}) = (\begin{bmatrix} \boldsymbol{f} \\ \boldsymbol{u} \end{bmatrix} \mid 0, \begin{bmatrix} K_{\boldsymbol{XX}} & K_{\boldsymbol{XZ}} \\ K_{\boldsymbol{ZX}} & K_{\boldsymbol{ZZ}} \end{bmatrix}), \tag{2}$$

where $\boldsymbol{f} = (f(\boldsymbol{x}_1), \ldots, f(\boldsymbol{x}_t))^T$. A variational distribution $q(\boldsymbol{u}) = \mathcal{N}(\boldsymbol{u} \mid \boldsymbol{m}, \boldsymbol{S})$ is used to approximate the posterior over inducing variables using the exact conditional distribution of $\boldsymbol{f}$ given $\boldsymbol{u}$, that is, $q(\boldsymbol{f}, \boldsymbol{u}) = p(\boldsymbol{f} \mid \boldsymbol{u})q(\boldsymbol{u})$. The posterior of $\boldsymbol{f}$ can be computed by marginalizing $\boldsymbol{u}$ with analytic form:

$$q(\boldsymbol{f}) = \int p(\boldsymbol{f} \mid \boldsymbol{u})q(\boldsymbol{u})d\boldsymbol{u} = \mathcal{N}(\boldsymbol{f} \mid \boldsymbol{Am}, K_{\boldsymbol{XX}} - \boldsymbol{A}^T(K_{\boldsymbol{ZZ}} - \boldsymbol{S})\boldsymbol{A}), \tag{3}$$

where $\boldsymbol{A} = K_{\boldsymbol{ZZ}}^{-1}K_{\boldsymbol{ZX}}$. The variational parameters $\boldsymbol{Z}, \boldsymbol{m}, \boldsymbol{S}$ are optimized by maximizing the Evidence Lower Bound (ELBO) which can be written in the following formulation[25]:

$$\mathcal{L}_1 = \sum_{i=1}^{t} \mathbb{E}_{q(f(\boldsymbol{x}_i))}[\log p(y_i \mid f(\boldsymbol{x}_i))] - \mathrm{KL}[q(\boldsymbol{u}) \parallel p(\boldsymbol{u})] = \mathcal{L}_{\mathrm{LL}} + \mathcal{L}_{\mathrm{KL}} \tag{4}$$

where $\mathrm{KL}[\cdot \parallel \cdot]$ is the KL divergence between two distributions. The ELBO breaks into a data likelihood term which factorized over training data, and a KL divergence term which can be computed in closed form. The factorization over data allows optimization via stochastic gradient descent (SGD), reducing the computational complexity to $\mathcal{O}(m^3)$.

## 4   Focalized Gaussian Process for Bayesian Optimization

Prior studies about variational sparse GPs are mainly designed for regression tasks, where the goal is to fit global training data distribution. In Bayesian optimization, the next sample is determined by the predictive function distribution over test positions. Gradient-based and evolutionary-based acquisition function optimization methods employ local search from random starting points to find a local optimal of the acquisition function. Recent works also scale grid search-based optimization to high dimensional space by restricting the search space within local sub-regions[46, 47]. All the above procedure would benefit from an accurate estimation over sub-region of the input space. Therefore, a sensible way to improve BO performance is to allocate limited computational resources to obtain better prediction over specific search regions instead of the entire input domain.

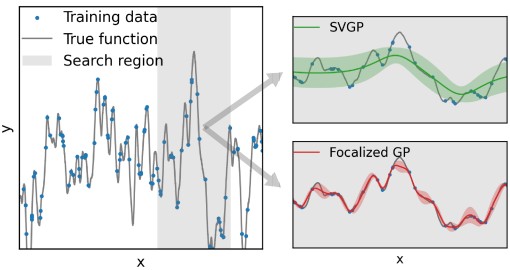

Figure 1: Performance comparison of focalized GP and SVGP over 1d GP functions. Posteriors are shown as mean $\pm$ 1 standard deviation.

We define the search region as the region where the acquisition function is optimized on, which is an axis-aligned hypercube with length $\boldsymbol{l} = (l_1, \cdots, l_d)^T$ centered at $\boldsymbol{c}$:

$$\mathcal{S}_{\boldsymbol{c}, \boldsymbol{l}} = \{\boldsymbol{x} \mid \boldsymbol{c} - \frac{1}{2}\boldsymbol{l} \le \boldsymbol{x} \le \boldsymbol{c} + \frac{1}{2}\boldsymbol{l}\}. \tag{5}$$

When $\boldsymbol{l} = (1, \cdots, 1)^T$ and $\boldsymbol{c} = (0.5, \cdots, 0.5)^T$, the acquisition optimization is performed over the entire input space $\mathcal{X}$, as commonly-used in vanilla BO algorithms. In the rest of this section, we first present the derivation of focalized loss function to improve GP prediction over the search region. Then we demonstrate how to incorporate our proposed GP model into Bayesian optimization.

## 4.1 Focalized evidence lower bound

We recall eq.1 and rewrite the mean estimation $\mu_t(\boldsymbol{x}_*)$ and variance estimation $\sigma_t(\boldsymbol{x}_*)$ for each test position $\boldsymbol{x}_*$:

$$\mu_t(\boldsymbol{x}_*) = \sum_{i=1}^{t} k(\boldsymbol{x}_*, \boldsymbol{x}_i)[K_{\boldsymbol{XX}} + \sigma I]^{-1} y_i,$$

$$\sigma_t(\boldsymbol{x}_*) = k(\boldsymbol{x}_*, \boldsymbol{x}_*) - \sum_{i=1}^{t}\sum_{j=1}^{t} k(\boldsymbol{x}_*, \boldsymbol{x}_i)\bar{k}_{ij}k(\boldsymbol{x}_*, \boldsymbol{x}_j), \tag{6}$$

where $\bar{k}_{ij}$ is the $(i,j)$-th entry of $[K_{\boldsymbol{XX}} + \sigma I]^{-1}$. From eq. 6 we can observe that the mean estimation at $\boldsymbol{x}_*$ is a linear combination of observation $\boldsymbol{y}$ multiplied by $k(\boldsymbol{x}_*, \boldsymbol{x})$, and the reduction of variance is a quadratic form of the covariance between $\boldsymbol{x}_*$ and training points. Both estimation can be written as linear summations of constant values with kernel function as weight. As mentioned in prior works[57], data points far from the test positions have a vanishingly small influence on the predictive distribution with commonly used kernel functions. Utilizing this observation, we propose to weight the data likelihood term using the kernel function to focus training over points that contribute to the prediction of the search region:

$$\mathcal{L}_{\text{WLL}} = \sum_{i=1}^{t} w_i \mathbb{E}_{q(f(\boldsymbol{x}_i))}[\log p(y_i \mid f(\boldsymbol{x}_i))], \quad w_i = \max_{\boldsymbol{x}_* \in \mathcal{S}_{\boldsymbol{c},\boldsymbol{l}}} k(\boldsymbol{x}_i, \boldsymbol{x}_*). \tag{7}$$

We use the maximum covariance of $\boldsymbol{x}_i$ to positions in the search region as the corresponding weight to filter out points that have marginally influence to the search region during GP training. In this way, the model can selectively utilize the training data to achieve good local prediction. When using a popular kernel functions such as RBF or Matern kernel, the maximum kernel value is equivalent to finding the nearest point in the search region, which can be easily calculated when the region boundary is axis-aligned as defined in eq. 5.

We additionally regularize the sum of weights to make the model focus on improving prediction over search region:

$$\mathcal{L}_{\text{reg}} = \frac{|\boldsymbol{X} \notin \mathcal{S}_{\boldsymbol{c},\boldsymbol{l}}|}{|\boldsymbol{X} \in \mathcal{S}_{\boldsymbol{c},\boldsymbol{l}}|} = \left(\frac{\sum_{i=1}^{t} w_i}{|\boldsymbol{X} \in \mathcal{S}_{\boldsymbol{c},\boldsymbol{l}}|} - 1\right), \tag{8}$$

where $|\boldsymbol{X} \in \mathcal{S}_{\boldsymbol{c},\boldsymbol{l}}| = \sum_{i=1}^{t} \mathbb{1}_{\boldsymbol{x}_i \in \mathcal{S}_{\boldsymbol{c},\boldsymbol{l}}}$ is the number of training points in the search region. The proposed regularization term $\mathcal{L}_{\text{reg}}$ encourages accurate local prediction instead of blurred global estimation, avoiding getting stuck on suboptimal of large kernel lengthscale. Combined with KL loss, our finalized new ELBO is as follows:

$$\mathcal{L}_2 = \mathcal{L}_{\text{WLL}} + \mathcal{L}_{\text{KL}} - \mathcal{L}_{\text{reg}}. \tag{9}$$

Compared to the original ELBO loss in SVGP, our proposed function maintains the same computational complexity and does not introduce additional hyperparameters. Our ELBO also reproduces eq. 4 when considering to predict the entire input space $\mathcal{X}$. During the model training, both GP hyperparameters and variational parameters are jointly optimized to obtain focalized GP for Bayesian optimization. Figure 1 shows a comparison of focalized GP and SVGP over 1d functions sampled from GP. While SVGP can only able to vaguely predict the function, focalized GP accurately delineate the function landscape within search region by training with the focalized loss. Our proposed GP model is sensitive to high-performing positions within the search space which contribute to better acquisition optimization. We also systematically compare the GP prediction performance in Appendix B.3, where our GP model trained from focalized ELBO consistently achieves good prediction on small size of search space.

**Theoretical implications of focalized ELBO.** Our focalized ELBO can be interpreted as a soft variant of training a local approximation over datapoints that lie within the search region. Here, we illustrate how local approximations can substantially reduce the KL divergence of the approximate posterior over the search region, and discuss the effects of tighter

approximations on BO regret bounds. We focus on providing general theoretical intuition rather than deriving precise bounds due to the lack of existing convergence guarantees for ELBO maximization in the general setting.

Suppose that we know the optimal point lies in some small sub-region of $\boldsymbol{X}$ that contains $N' << N$ training points. Corollary 19 in [58] shows that given a squared exponential kernel and some assumptions on the inducing point selection, for a fixed number of inducing points the KL-divergence upper bound scales super-quadratically in the number of training points. Hence, fitting locally can yield much tighter approximations than fitting globally (e.g. SVGP).

Next, we consider the impact of the KL approximation error on the optimization regret. Proposition 1 in [58] states that the gap between the means of the approximate and exact posteriors is upper bounded by $\mathcal{O}(\sigma\sqrt{\gamma})$, where $\gamma$ is an upper-bound on the approximation KL-divergence. This has an immediate impact on the regret - for example, when GP-UCB [51] is combined with sparse GPs, the confidence bounds must be enlarged by an additive $\sqrt{\gamma}$ factor to account for the approximation error. Because the regret bound scales with $\sqrt{\beta_T}$ where $\beta_T$ is the maximum confidence interval coefficient, having a large approximation error can arbitrarily scale the regret incurred by the algorithm. In order to achieve no additional regret order, the additional approximation error noise must be uniformly bounded (Assumption 4 in [37]). Although focalized GP cannot guarantee a constant bound, it still directly reduces the regret of the algorithm, where we empirically investigate in Appendix B.1.

## 4.2 Bayesian optimization with focalized GP

One advantage of focalized GP is that it can be easily integrated into existing BO algorithms. To further leverage the strong local modeling properties of focalized GP, we design `FocalBO`, a hierachical acquisition optimization framework described in Algorithm 1.

At each BO iteration, `FocalBO` iteratively optimizes the acquisition function over a progressively smaller search region via focalized acquisition function (`FocalAcq`) as shown in Algorithm 2. The first depth of acquisition optimization starts with the entire input space $\mathcal{X}$ with $\boldsymbol{l} = (1, \cdots, 1)^T$ and $\boldsymbol{c} = (0.5, \cdots, 0.5)^T$ (line 1). We train specific focalized GP base on the search region at each round of acquisition optimization (line 4-5). Our framework is compatible with any acquisition function that extracts instant posterior information from the GP and is optimized within pre-defined search region. After one round of acquisition function optimization, the search space length $\boldsymbol{l}$ is halved to focus on a smaller search region centered at current best

---

**Algorithm 1 FocalBO**

**Input** Initial Dataset $\mathcal{D}_0$, Inducing Variable Size $m$, Batch Size $B$

1:    $H \leftarrow 1$
2:    **for** $t = 1, 2, \cdots$ **do**
3:      $\{\boldsymbol{x}_{t,i}\}_{i=1}^B, \{h_{t,i}\}_{i=1}^B \qquad \leftarrow$ FocalAcq$(\mathcal{D}_{t-1}, H, m, B)$
4:      Observe $\{y_{t,i}\}_{i=1}^B = \{f(\boldsymbol{x}_{t,i}) + \eta\}_{i=1}^B$

5:      $\mathcal{D}_{t+1} \leftarrow \mathcal{D}_t \cup \{(\boldsymbol{x}_{t,i}, y_{t,i})\}_{i=1}^B$
6:      $i_{\text{best}} \leftarrow \text{argmax}_{i \in 1, \cdots, B} y_i$
7:      **if** $h_{i_{\text{best}}} < H$ **then**
8:        $H \leftarrow H - 1$
9:      **else**
         $H \leftarrow H + 1$
10:     **end if**
11:   **end for**

---

position $\boldsymbol{x}_{\text{best}}$(line 6-7). In this way we can obtain a more accurate model for decision making, and also relieve the over-exploration problem when the problem dimension is high[59]. One batch of inputs is proposed at each round of optimization, and the final decision is sampled from all proposed inputs via Softmax distribution over their corresponding acquisition function values (line 9). Our hierarchical optimization strategy enables collecting candidates from both global sparse estimation and local focalized prediction, achieving balance between exploration and exploitation with constrained computation power.

The optimization depth $H$ in `FocalAcq` controls the degree of utilizing local information from current best position, where the GP estimate variance decreases with the shrinkage of search space. The best-performing optimization depth is likely problem-dependent (e.g. high-dimensional functions may require higher optimization depths). Therefore in `FocalBO`, we

propose to automatically adjust the optimization depth according to the instant optimization performance. At the beginning of the optimization, we initialize the optimization depth as 1, indicating global search of the input space (Algorithm 1, line 1). Then we keep track of the depth where the proposed positions are sampled from. If the depth of the best point in this round is less than the current optimization depth $H$, we reduce $H$ to encourage exploration of the input space, otherwise we increase $H$ for better exploitation of $\boldsymbol{x}_{\text{best}}$ (line 6-10).

Our proposed framework is orthogonal to TuRBO-M [46], but bears some similarities in searching over multiple sub-regions and adaptively adjusting the search region. Our algorithm differs in that TuRBO-M constructs equal-sized trust regions and fits independent Exact GP using separated dataset, aiming at searching for different local optima in the search space. By contrast, the search region in FocalBO is constructed with different sizes to make decision based on both global and local information. Our framework allows data sharing across search regions, and the use of focalized GP helps to accurately estimate local region with limited representation. Additionally, FocalBO does not introduce extra hyperparameters. Finally, we demonstrate in Section 5 that TuRBO is complementary to FocalBO in optimizing high-dimensional problems.

---

**Algorithm 2** FocalAcq

---

**Input** Dataset $\mathcal{D}_{t-1}$, Optimization Depth $H$, Inducing Variable Size $m$, Batch Size $B$
1: $\boldsymbol{l} \leftarrow (1, \cdots, 1)^T, \boldsymbol{c} \leftarrow (0.5, \cdots, 0.5)^T$
2: Select current best point $\boldsymbol{x}_{\text{best}}$ from $\mathcal{D}_{t-1}$
3: **for** $h = 1, \cdots, H$ **do**
4:     Train $\mathcal{M}_t^h$ using $\mathcal{L}_2$ given $\mathcal{S}_{\boldsymbol{c},\boldsymbol{l}}$
5:     $\{\boldsymbol{x}_{t,i}^h\}_{i=1}^B \leftarrow \text{argmax}_{\boldsymbol{x} \in \mathcal{X}_*} a(\boldsymbol{x}|\mathcal{M}_t^h)$
6:     $\boldsymbol{l} \leftarrow \boldsymbol{l}/2$
7:     $\boldsymbol{c} \leftarrow \boldsymbol{x}_{\text{best}}$
8: **end for**
9: **return** $\{\boldsymbol{x}_{t,i}\}_{i=1}^B, \{h_{t,i}\}_{i=1}^B \sim P(i = i') \propto$
$$\frac{\exp^{a(\boldsymbol{x}_{t,i'}^{h'}|\mathcal{M}_t^{h'})}}{\sum_{h=1}^H \sum_{j=1}^B \exp^{a(\boldsymbol{x}_{t,j}^h|\mathcal{M}_t^h)}}$$

---

## 5 Experiments

In this section, we extensively evaluate FocalBO over a variety of tasks. We first use synthetic functions to showcase the compatibility of FocalBO in improving commonly-used acquisition functions. Next, we consider the online optimization of robot morphology design that is additionally given a large offline dataset. We also show that FocalBO is able to optimize very high-dimensional musculoskeletal system control with both a large offline dataset and a large number of online budget. Finally we dig deeper into FocalBO to analyze how each of its components contributes to superior optimization performance.

We compare FocalBO with representative sparse GP models used for Bayesian optimization, including SVGP[25], WOGP[10], and Vecchia GP[12]. We only run WOGP on synthetic functions due to its extremely low speed in dealing with the datasets in the remaining tasks. The number of inducing variables in sparse GP models is set as 50 for synthetic functions and as 200 for other tasks. The optimization performances are shown as mean $\pm$ 1 standard error for all considered problems over 10 independent trials.

### 5.1 Synthetic functions

We select Shekel and Michalewicz as the test functions, which are heterogeneous with both smooth and rigid regions. We also sample functions directly from Gaussian processes to evaluate algorithm performance under full BO assumption. For each function, we choose to use different acquisition functions to optimize: TS optimized by grid search, EI optimized by analytic gradient, and probability of improvement (PI) optimized by Monte Carlo gradient[55]. Optimization performances are shown in Figure 2. We observe that FocalBO significantly improves the performance of all acquisition functions compared to SVGP, and is able to consistently achieve top-tier performance over all problems. In Michalewicz function where a large fraction of the input space is flat, all baselines tend to increase the noise estimation to maintain a stationary prediction, while focalized GP is able to focus on the local search region and successfully optimize the function. Additional experiment with online samples as major data source is shown in Appendix B.2, where FocalBO still maintains comparable or better performance against baselines.

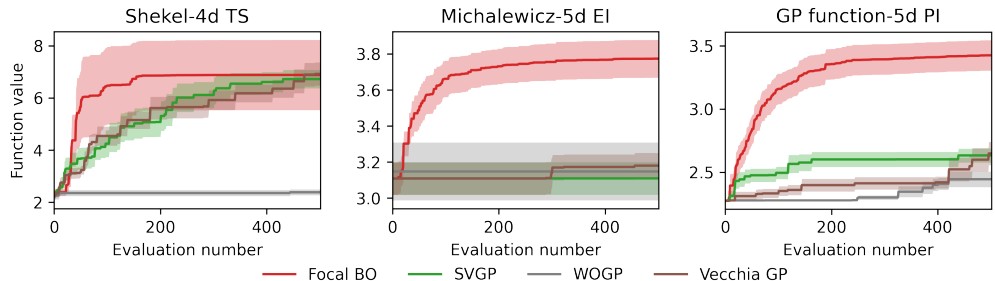

Figure 2: Optimization performance under different synthetic function and acquisition function. Sparse GP models are trained with 50 inducing variables. The offline dataset contains 2000 random data points and the online budget is 500 with batch size of 10.

## 5.2 Robot morphology design

We compare FocalBO to several baselines over robot morphology design task from Design-Bench, which provides large offline dataset with an exact function oracle[6]. The goal of the task is to optimize the morphological structure of D'Kitty robot[60] to improve the simulation performance under RL controller. While the benchmark is initially designed for offline model-based optimization (MBO), it can also be used as an offline-to-online BO benchmark. In this task, we use the training dataset with 10,000 points and additionally evaluate 128 points on-the-fly with batch size of 4. EI is used as the base acquisition function for better optimizing with small batch size. We also try to combine FocalBO with TuRBO to optimize over the high-dimensional space, with the results shown in Figure 3. We observe that FocalBO achieves significant improvement from the initial data while other baselines struggle to obtain performance gain, even combined with TuRBO. FocalBO with TuRBO effectively extracts information from large offline dataset and is the first GP-based method to achieve top-tier performance reported by prior MBO works[61].

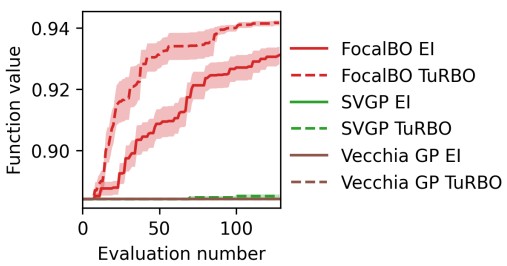

Figure 3: Optimization on robot morphology design. Function values are normalized by best and worst values in the unseen full dataset.

## 5.3 Human musculoskeletal system control

We further apply FocalBO to control a human arm musculoskeletal system[62] for the task of pouring liquid into a cup, as shown in Figure 4(a). To control the musculoskeletal system, we optimize a linear policy $\pi \in |A| \times |O|$, where $|A| = 5$ and $|O| = 117$ are the corresponding action and observation dimensions. The action dimension has been reduced from individual muscles to synergetic groups of muscles by applying principled component analysis to sampled action data from an RL agent (Appendix A.6). Although the original control dimension is reduced, the remaining 585-dimensional input space is still very high for existing high-dimensional BO algorithms. Therefore we consider a large offline-online setting, where we randomly sample 2000 points from the input space to serve as the offline dataset, and set the online budget as 3000 with batch size of 100. We use Thompson sampling as the base acquisition function. Figure 4(b) demonstrates that FocalBO outperforms other baselines, achieving higher maximum reward and faster convergence speed. Our supplementary video shows that the optimized policy is able to perform well on the task, demonstrating the successful application of FocalBO to high-dimensional control problems.

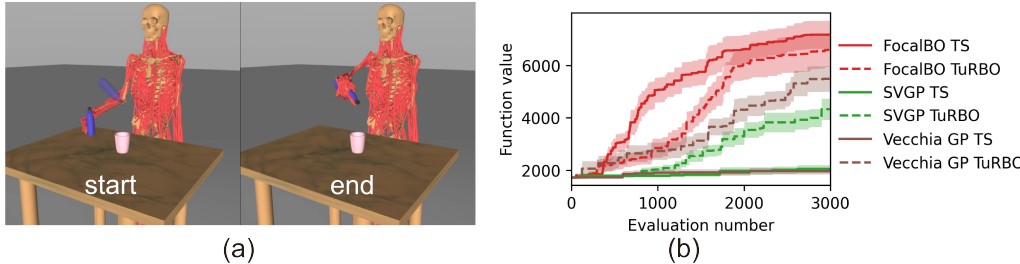

(a)          (b)

Figure 4: Optimization of musculoskeletal system control. (a) Task illustration of initial and target state. Full video in supplementary. (b) Optimization performance of algorithms.

## 5.4 Algorithm analysis

To understand the reasons behind FocalBO's superior optimization performance, we investigate the optimization depth in FocalBO, which is the central component of the method. Figure 5(a) shows the evolution of optimization depth over different problems, where FocalBO is able to adapt the optimization depth according to different function structure. For Shekel and musculoskeletal model control where the promising regions are distinct, the optimization exhibits an increasing trend to exploit current best points, while for other problems the depth tends to converge at a fixed level. Figure 5(b) shows the sources of proposed batches during the optimization of musculoskeletal system control. Overall the samples exhibits clear trend from exploration to exploitation over high-dimensional input space. Our hierarchical optimization strategy enables flexibility between exploration and exploitation.

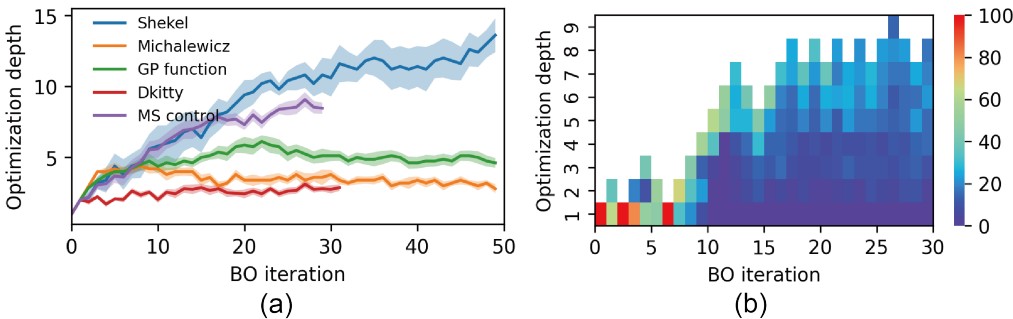

(a)          (b)

Figure 5: Algorithm analysis over optimization depth. (a) Depth evolution during optimization. (b) Samples source of each BO iteration during one trial of musculoskeletal system control optimization. Color bar indicates the number of samples proposed by corresponding optimization depth.

## 6 Conclusion

In this paper, we propose FocalBO, which uses a hierarchical acquisition optimization strategy equipped with focalized GP model to scale Bayesian optimization to problems with large offline datasets and/or a large number of online samples. Despite limited representation capability, FocalBO consistently improves various acquisition functions in optimizing heterogeneous functions, and adeptly leverages large offline dataset for efficient optimization over robot morphology. Under the large offline-to-online optimization setting, FocalBO achieves stable high-dimensional control of human musculoskeletal model with over 500 parameters. Ablation studies over the algorithm components further verify the principled design of FocalBO. Future work may include theoretically analyzing FocalBO, and applying the method to more complex problems, such as large-scale parameter tuning and whole-body human musculoskeletal system control.

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

# A Implementation Details

## A.1 Implementation of FocalBO

We implement FocalBO with BoTorch[1], which is a popular library for BO implementation with GPU acceleration. For acquisition optimization, we directly use acquisition function implementation and corresponding optimizers from BoTorch. Our code for fully reproducing all experimental results is in the: `https://github.com/yunyuewei/FocalBO`. Our musculoskeletal model will be released soon. In the meantime, the model can be accessed for research purposes upon request (ysui@tsinghua.edu.cn).

## A.2 Implementation of baselines

**SVGP**. We directly use approximated GP class in Gpytorch example[2].

**WOGP**. We refer to the original implementation[3], and write a Botorch GP wrapper with inducing point kernel to enable acquisition optimization using BoTorch. As the hyperparameter are unknown to the GP model, we first warm up WOGP using random set of inducing points for 100 epochs, then perform weighted training point selection and continue hyperparameter fitting with the selected WOGP model.

**Vecchia GP**. We directly use the original implementation[4] without much modification, as it is also implemented in BoTorch.

**TuRBO**. We refer to the implementation in BoTorch tutorials[5], and use the default setting in trust region length and success/failure thresholds.

## A.3 GP training details

For all GP, we use Matern $\frac{5}{2}$ kernel with automatic relevance determination, and do not restrict the lengthscale or noise range. For each round of GP training, we fit GP hyperparameters (and variational parameters for focalized GP and SVGP) for 1000 epochs via Adam optimizer[63] with learning rate as 0.01. For focalized GP and SVGP, we initialize the inducing points using Sobol sampler[64] over input space. all experiment are conducted on a server with Intel(R) Xeon(R) Gold 6348 CPU @ 2.60GHz, NVIDIA-A100 and 512Gb memory.

## A.4 Synthetic functions

For GP function, we directly sample from a exact 5d GP using Matern $\frac{5}{2}$ kernel with lengthscale as 0.5. For other synthetic functions, we directly use the test function implementation from BoTorch.

## A.5 Robot morphology design

We use the dataset and function oracle from Design Bench[6]. We choose D'Kitty morphology design for its consistency in function values between offline dataset and online function oracle, and its compatibility with python 3.8+.

## A.6 Human musculoskeletal system control

We use the musculoskeletal system from [62], which enables foward simlation with Mujoco[65] and environment customization. We design the following reward for each environment step:

---

[1]`https://botorch.org/`
[2]`https://gpytorch.ai/`
[3]`https://github.com/ermongroup/bayes-opt`
[4]`https://github.com/feji3769/VecchiaBO/tree/master/code/pyvecch`
[5]`https://botorch.org/tutorials/turbo_1`
[6]`https://github.com/brandontrabucco/design-bench`

$$r = 50r_{\text{pos}} * 10r_{\text{ori}} + 10r_{\text{reach}} + r_{\text{lift}} - r_{\text{act}} - 5r_{\text{done}} \tag{10}$$

where $r_{\text{pos}}$ encourages the bottle near the target position, $r_{\text{ori}}$ encourages the bottle near the target orientation, $r_{\text{reach}}$ encourages the hand to grab the bottle, $r_{\text{lift}}$ encourages the hand to lift the bottle, $r_{\text{act}}$ penalize the overall muscle activation, $r_{\text{done}}$ penalize the early ended episode due to dropped bottle or hand outside of pre-defined range.

We trained a Soft Actor-Critic (SAV) [66] agent for 6M timesteps to collect task-related muscle activation data, and use principled component analysis to reduce the action dimension from 81 to 5.

# B    Additional Experiments

## B.1    Theoretical implications of sparse GP approximation.

In Figure 6, we also empirically measure our claim that Focalized GP can significantly reduce approximation error on the search region. We sampled 8000 training points from 2d GP functions to train focalized GP and SVGP. Over different size of the search region, we compare the KL divergence of the GP posterior prediction over search region between sparse GPs and the exact GP. We observe that the KL divergence between focalized GP and exact GP is consistently smaller than that between SVGP and exact GP, implying tighter approximation to the exact GP over local region.

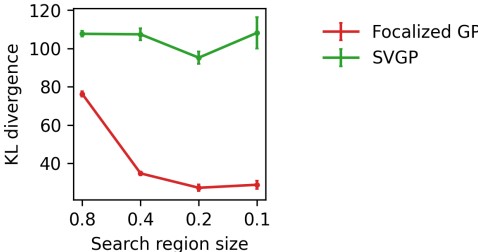

Figure 6: KL divergence between sparse GPs and exact GP. Results shows the mean and one standard error, averaged over 50 independent trials.

While a rigorous regret bound is hard to derive, we conduct an empirical study where we directly compare the optimization performance between focalized GP and SVGP when combining with TuRBO. In this way we can eliminate the influence of hierarchical acquisition optimization. The optimization performances are shown in Figure 7. We observe that focalized GP outperforms SVGP on both high-dimensional problems, which empirically demonstrates our theoretical implications that Focalized GP contributes to reducing regret.

**Different way of centering the search region**

We empirically investigate this in Figure 8 (a), which compares different ways of selecting the search region center by measuring the distance from the search region center to the global optima. We observe that current best point consistently is the closest to the global optimum, which validates this design choice.

For the experiment above, we sampled 2d functions from GPs with Matern $\frac{5}{2}$ kernel and lengthscale of 0.05 (representing rigid functions), and selected the best point over unifromly sampled 10,000 points as the global optima.

A sparse GP is already more explorative than using the full GP, since the smaller representational capacity leads to smoother posteriors. In Figure 8 (b), We demonstrate this empirically below, where we measure the pair-wise distance of 100 Thompson sampling points under exact and SVGP (with 50 inducing points). We observe that sparse GP actually samples more diverse sets compared to exact GPs, i.e. exhibiting more exploration. Therefore,

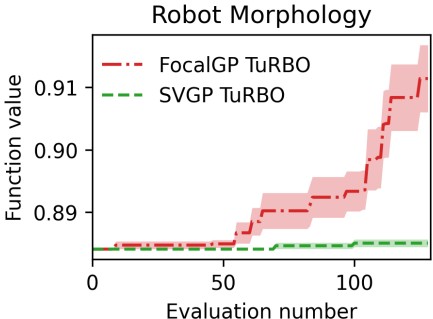
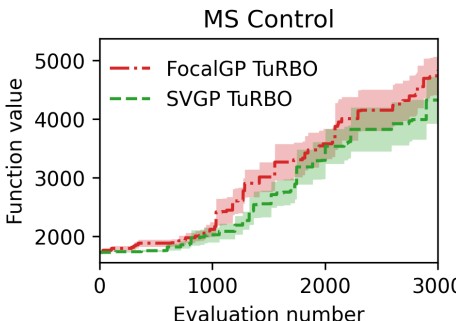

Figure 7: Optimization performance of focalized GP and SVGP when combining with TuRBO.

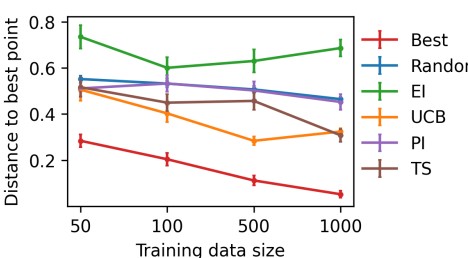
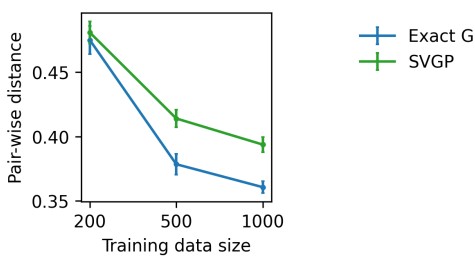

Figure 8: (a) Distance of search region center to the global optima. (b) Pair-wise distance of Thompson sampling samples. Results shows the mean and one standard error, averaged over 50 independent trials.

using focalized GPs does not sacrifice exploration, and significantly helps exploitation by performing acquisition function optimization over smaller search regions.

### B.2  Optimization on synthetic functions with large online data

We choose Ackley and Hartmann, which are common-used test functions for BO community. We use the similar optimization setting in [12]. The optimization performances are shown in Figure 9, where FocalBO is still able to achieve comparable or better performance when online samples dominates the data source.

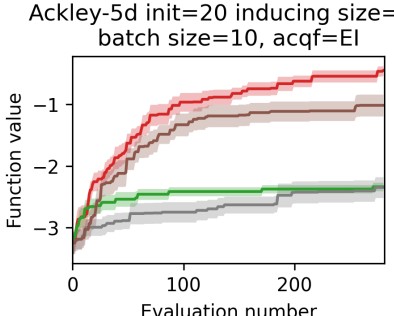
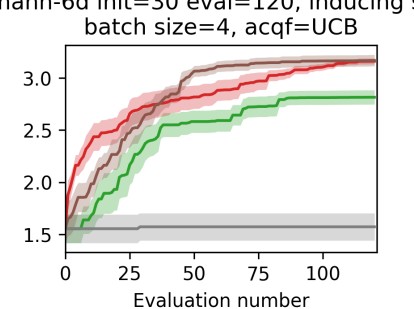

Figure 9: GP predictive performance of specific search region on 2d Ackley and Rastrigin function. Results show mean ± one standard deviation over 10 random search regions.

## B.3 GP predictive performance

We use two common-used synthetic functions, Ackley and Rastrigin, to analyze the the GP predictive performance of focalized GP compared with Exact GP and SVGP under different search region size $l$ and different inducing variables number $m$. We show the negative log likelihood (NLL) and root mean squared error (RMSE) in Figure 10. The results shows that focalized GP outperforms both Exact GP and SVGP in terms of both NLL and MSE when the search space size is lower than 0.5. In Rastrigin function where Exact GP achieves similar performance as SVGP, focalized GP is still able to accurately predict the local search region over different choice of inducing variable numbers. We also show in Figure 11 that the regularization term $\mathcal{L}_{\text{reg}}$ is indispensable to the training of focalized GP to achieve good local prediction.

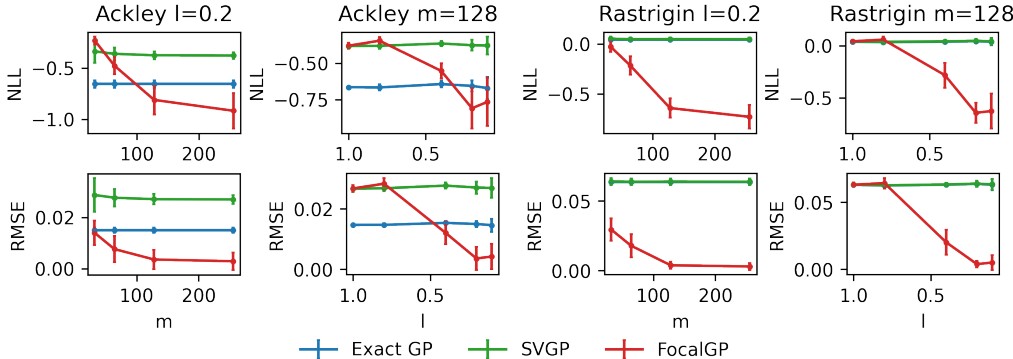

Figure 10: GP predictive performance of specific search region on 2d Ackley and Rastrigin function. Results show mean ± one standard deviation over 10 random search regions.

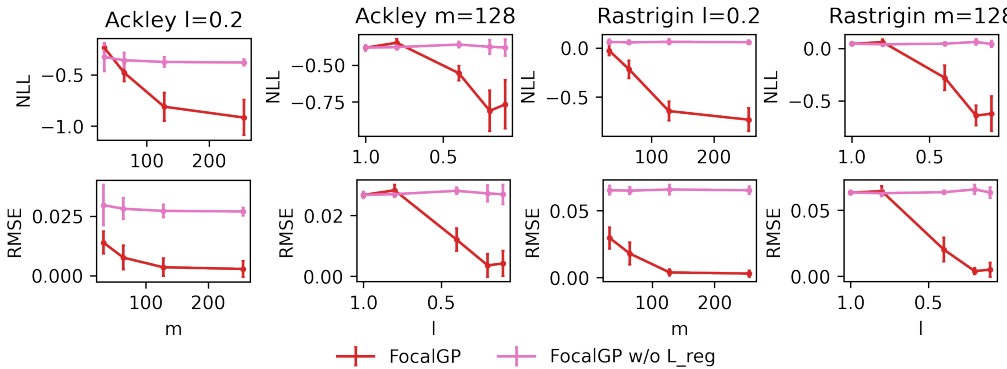

Figure 11: Ablations on the regularization loss $\mathcal{L}_{reg}$. Results show mean ± one standard deviation over 10 random search regions.

## B.4 Comparison with TuRBO

We run the original TuRBO implementation (with exact GP and Thompson sampling) and TuRBO with nearest neighbor GO model on both robot morphology design and human musculoskeletal system control task (Figure 12). We observed that FocalBO outperforms TuRBO on both tasks with smaller computational cost. The reason of TuRBO's poor performance may be that it cannot quickly adapt over the search space when the online evaluation budget is small.

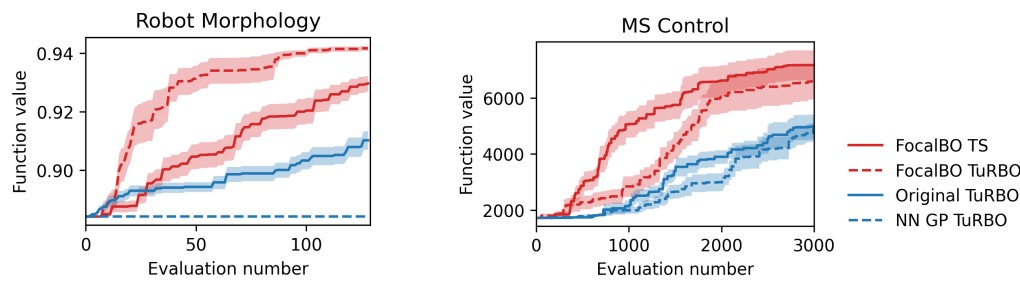

Figure 12: Optimization performance of FocalBO and TuRBO.

# C    Lemmas Used for Theoretical Implications of Focalized ELBO

**Lemma 1.** *(Corollary 19 in [58]). Let k be a squared exponential kernel. Suppose that N real-valued (onedimensional) covariates are observed, with identical Gaussian marginal distributions. Suppose the conditions of Theorem 13 are satisfied for some $R > 0$. Fix any $\gamma \in (0, 1]$. Then there exists an $M = \mathcal{O}(\log(N^3/\gamma))$ and an $\epsilon = \Theta(\gamma/N^2)$ such if inducing points are distributed according to an $\epsilon$-approximate M-DPP with kernel matrix $K_{ff}$,*

**Lemma 2.** *(Proposition 1 in [58]). Suppose $2KL[Q \parallel P] \leq \gamma \leq \frac{1}{5}$. For any $x^* \in \mathcal{X}$, let $\mu_1$ denote the posterior mean of the variational approximation at $x^*$ and $\mu_2$ denote the mean of the exact posterior at $x^*$. Similarly, let $\sigma_1^2, \sigma_2^2$ denote the variances of the approximate and exact posteriors at $x^*$. Then,*

$$|\mu_1 - \mu_2| \leq \sigma_2\sqrt{\gamma} \leq \frac{\sigma_1\sqrt{\gamma}}{\sqrt{1 - \sqrt{3\gamma}}} and |1 - \sigma_1^2/\sigma_2^2| < \sqrt{3\gamma} \tag{11}$$

**Lemma 3.** *(Assumption 4 in [37]). (quality of the approximate prediction). For the approximate $\tilde{\mu}_t$, the exact $\mu_t$ and $\sigma_t$, and for all $x \in \mathcal{X}$,*

$$|\tilde{\mu}_t(x) - \mu_t(x)| \leq c_t\sigma_t(x), \tag{12}$$

*where $0 \leq c_t \leq c$ for all $t > 1$ and some constant $c \in \mathbb{R}$*

