# OpenReview forum: "Scalable Bayesian Optimization via Focalized Sparse Gaussian Processes"
_NeurIPS.cc/2024/Conference — NeurIPS 2024 poster_

### Official Review · Reviewer_BXny · 2024-07-08

**Soundness:** 3
**Presentation:** 3
**Contribution:** 3
**Rating:** 7
**Confidence:** 3

**Summary:**

The paper presents a new method for scaling Bayesian Optimisation to large datasets by employing sparse GP in a way that facilitates BO.  The proposed method of Focal BO fits a sparse GP at different scales, optimises the acquisition function for each of them and selects the next point to query by sampling with probability proportional to its acquisition function value at its scale. The proposed algorithm is evaluated at a number of benchmarks, where a large number of data points is available.

**Strengths:**

The paper addresses a very important and, in my opinion, understudied topic of scaling BO to large datasets. The proposed algorithm is elegant and does not introduce any new hyperparameters. It also seems to be delivering a significant improvement in performance over baselines.

**Weaknesses:**

- Since the paper is mostly empirical I would expect a bit more baselines. The simplest way to scale TuRBO to large datasets is simply to keep $N$ points closest to the optimal solution. It would be great to see how this naive "keep closest N TuRBO" baseline compares to FocalBO TuRBO.

- "FocalBO with TuRBO (...) is the first GP-based method to achieve top-tier performance reported by prior MBO works." - this is a strong statement without much justification. Please either directly compare with MBO baselines or quote the exact numbers achieved by prior work on the exact same setting as considered in the paper.

**Questions:**

-  In Equation 8, why is there a $ - 1$ at the end? Since it is a constant it should not affect the optimisation process at all.

- I cannot understand why authors refer to ELBO as "ELBO Loss", since due to how it is defined, I imagine you would rather want to maximise it (as it corresponds to expected likelihood). Are you maximising the loss function? In that case, I do not think it is proper to call it that name.

**Limitations:**

Within the paper authors do not discuss the limitations (or at least I cannot find them).

---

> ### Author Rebuttal · Authors · 2024-08-07
>
> Thanks for your appreciations of our problem setting and algorithm design. We address your concerns below. Please find the rebuttal Figures by opening the rebuttal PDF file.
>
> **Baselines comparison**
>
> In **Figure R4**, We run the "keep closest N TuRBO" baseline (denoted as NN GP TuRBO) on both robot morphology design and human musculoskeletal system control task, and find that FocalBO consistently outperforms this simple baseline. Thanks to reviewer SUei, we find that FocalBO also outperforms original TuRBO implementation on both high-dimensional benchmarks. The reason of TuRBO's poor performance may be that it cannot quickly adapt over the search space when the online evaluation budget is small.
>
> In terms of the statement in section 5.2, we find that there is no statistically significant difference between FocalBO and the best performing algorithm reported in Design Bench baselines (p value=0.1168). We will extensively compare the optimization performances against MBO baselines in the new version.
>
> **The design of $\mathcal{L}_{\text{reg}}$**
>
> Indeed, the -1 does not affect the loss, and is primarily for illustrative purposes. Specifically, we can write
>
> $L_{\text{reg}} = \frac{
> \sum_{i=1}^t
>  w_i - |X \in S_{c,l}|
> }{
> |X \in S_{c,l}|
> } $
> $=
> \frac{
> \sum_{i=1}^t
>  w_i -\sum_{i=1}^t
>  1_{x\in S_{c,l}} w_i
> }{
> \sum_{i=1}^t 1_{x\in S_{c,l}} w_i
> }
> = \frac{
>  \sum_{i=1}^t
>  1_{x\notin S_{c,l}} w_i
> }{
> \sum_{i=1}^t 1_{x\in S_{c,l}} w_i
> }
> $
>
> which motivates the regularization term as minimizing the weights of points outside of the search region, while normalizing by the size of the search region (to account for decreasing search region sizes in FocalBO).
>
> **ELBO statements**
>
> Thanks to pointing out the confusing loss description - it's a typo on our end. We actually maximize the weighted data likelihood term and minimize the regularization term, therefore $\mathcal{L}_{\text{reg}}$ should be multiplied by $-1$. We will unify the definitions to the ELBO description to reduce confusion.
>
> **Limitations of our work**
>
> While FocalBO demonstrate superior performances against existing scalable BO algorithms, theoretical can be further improved to make it more rigorous. For instance, an analysis of the convergence of FocalBO would enhance our understanding of the algorithm’s mechanism. The currently-used hypercube search region may not align well with the underlying function landscape, potentially wasting computational resources on low-performing regions. A more sophisticated search region design such as [1] may help FocalBO to further improve the optimization performance.
>
> [1] Wang, Linnan, Rodrigo Fonseca, and Yuandong Tian. "Learning search space partition for black-box optimization using monte carlo tree search." Advances in Neural Information Processing Systems 33 (2020): 19511-19522.

---

> > ### Comment · Reviewer_BXny · 2024-08-11
> >
> > Thank you very much for your rebuttal. Given the new experiments, I am willing to increase my score to 7 (accept).
> >
> > Regarding the novelty issues mentioned by Reviewer SUei, I believe the TuRBO algorithm became universally accepted as the go-to solution when it comes to high-dimensional Bayesian Optimisation. As such, in my opinion, the fact that FocalBO is merged with TuRBO to produce a stronger baseline is not really a problem, given that it severely outperforms other TuRBO-based baselines. It is clear that the main component critical to success is the FocalBO strategy. I also believe that the FocalBO itself is sufficiently different from the previously proposed sparse-GP frameworks.

---

> > > ### Author Response · Authors · 2024-08-11
> > >
> > > Thank you for your feedback on the new experiments and the novelty of our work! We will improve the presentation of the manuscript by incorporating the new experimental results and highlight our novelty in the revised version.

---

### Official Review · Reviewer_SUei · 2024-07-12

**Soundness:** 3
**Presentation:** 3
**Contribution:** 2
**Rating:** 4
**Confidence:** 4

**Summary:**

Operating in the context of Bayesian optimization, the authors propose to train a surrogate model which focuses on a specific sub-region of the input space by weighting the log-likelihood contributions of each datapoint relative to their distance to that region. They also propose an algorithm for choosing this subregion over the course of optimization, which they demonstrate in numerical experiments.

**Strengths:**

The reasoning and motivation at a conceptual level is clear and the article is well-written at a high level.

I really like Section 5.4 which gives us an idea of how the algorithm is attacking the problem.

The test problems used in Section 5.2 and 5.3 of the numerical experiments are outstanding, as they represent nontrivial problems and are of high dimension.

**Weaknesses:**

The Lreg term of equation 8 seems to come out of left field, and the implications of adding it to the cost in equation 9 are insufficiently discussed; I say more about this in the Questions section.

Figure 1 is not super convincing: it's clear that with a different hyperparameter/inducing point setting (perhaps with admittedly more inducing points), that SVGP would be able to interpolate as well.

The theoretical implications section is underdeveloped, containing neither new math nor computational experiments verifying that the claims made there hold up empirically.

Regarding novelty level, I would assess it as medium as it falls within these existing frameworks of trust-region style optimization spawning from Erikkson et al as well as work about tailoring inducing point methods to consider specific parts of the input space (a different implementation but a similar spirit is [1]), though the combination to Bayesian optimization is in my view novel.

[1] Cole, D. Austin, Ryan B. Christianson, and Robert B. Gramacy. "Locally induced Gaussian processes for large-scale simulation experiments." Statistics and Computing 31.3 (2021): 33.


Only for the benefit of the authors, I am attaching here some grammar/clarity issues which did not affect my scoring of this article:
201) shows an comparison -> shows a comparison
232) directly reduces
246-248) sentence spanning these two lines needs attention.
251) from GP -> from the GP
298) both large offline dataset and large number -> both a large offline dataset and a large number
304) inducing variable number -> number of inducing variables
374) "consistently augments" does not make sense in context; I'm not sure what you're trying to say.

**Questions:**

I understand that Lreq (Eq 8) is given such that it is divided by the number of points in the subregion and has one subtracted to it such that its minimum possible value is zero, and since the w's for any points inside S is constant, we are effectively minimizing the sum of kernel functions evaluated at extra-S training points with the projection of that point onto S. Why is this the right thing to do? It's going to bias lengthscale estimates to be small relative to a GP fit to all the data or even one fit only to data inside the subregion. Why is this desirable, and why is Lreg the right way to operationalize this?

Am I correct that none of FocalBO Turbo, SVGP Turbo, or Vecchia TurBO represent the actual algorithm originally proposed my Eriksson et al, but rather are re-implementations of TuRBO's strategy in other approximate GP frameworks?

I think there are some interesting ideas here, but I regretfully am presently suggesting rejection as a result of the limited scope of the competitors in the numerical experiments which in my view is required of a methodology paper with this medium level of conceptual novelty appearing in Neurips. I think this is a solid paper but does not yet meet the high bar of a Neurips article.

**Limitations:**

You say that you don't introduce any additional hyperparameters, but it might be more accurate to say that you provide defaults for additional hyperparameters: you set the relative strength of Lreg to the inverse of the number of points in the search region (which makes sense for making the minimum of this objective zero, but it's not clear what implications the scaling has) and you have a specific heuristic for choosing the optimization depth. Why is it reasonable to believe this would still be the right thing to do on some other black-box test function outside of the five you've presented here?

I think the problems in Section 5.2 and 5.3 are really interesting problems.
But to help compare against prior work, it would be really important to compare against TuRBO itself: though TuRBO uses an "exact" GP, they use numerical analysis tricks to scale to large datasets; (glancing at the paper,they do 20,000 evaluations on a 60D problem). So it seems like TuRBO as originally presented by Eriksson et al should have been a competitor in this simulation study.

---

> ### Author Rebuttal · Authors · 2024-08-07
>
> Thank you for your detailed review. We are improving the paper based on your suggestions. We'd also like to clarify your concerns in the following paragraphs. Please find the rebuttal Figures by opening the rebuttal PDF file.
>
> **GP comparison in Figure 1**
>
> We would like to highlight that our goal of designing focalized GP is to to **allocate a fixed set of limited computational resources to obtain better prediction over specific search regions instead of the entire input domain**. This would contribute to better acquisition function optimization in BO as local optimization are usually employed (line 147-165 in the paper).
>
> For any function, there is some number of inducing points for which the sparse GP posterior converges to that of the full GP. In practice, however, this threshold is unknown, and may be prohibitively high for complex functions. Hence, Figure 1 attempts to illustrate that for a **fixed number of inducing points**, there is a failure mode in which attempting to model the full input space results in an overly-smooth posterior, and that one possible remedy (ours) is to allocate this representational budget towards a subregion of the input space.
>
> **The design of $\mathcal{L}_{\text{reg}}.$**
>
> Without this regularization term, we observed that the ELBO loss can lead to arbitrarily high weights, leading to large lengthscales and poor fidelity within the search region. Hence, we designed $\mathcal{L_{\text{reg}}}$ to emphasize prediction within the search region, since we are only optimizing within those subregions in FocalBO. To better illustrate the formulation of $\mathcal{L}_{\text{reg}} $, we can rewrite it as
>
>
>
> $L_{\text{reg}} = \frac{
> \sum_{i=1}^t
>  w_i - |X \in S_{c,l}|
> }{
> |X \in S_{c,l}|
> } $
> $=
> \frac{
> \sum_{i=1}^t
>  w_i -\sum_{i=1}^t
>  1_{x\in S_{c,l}} w_i
> }{
> \sum_{i=1}^t 1_{x\in S_{c,l}} w_i
> }
> = \frac{
>  \sum_{i=1}^t
>  1_{x\notin S_{c,l}} w_i
> }{
> \sum_{i=1}^t 1_{x\in S_{c,l}} w_i
> }
> $
>
> in which the numerator encourages reducing the weights of points outside search region (leading to better estimation within the search region), and the denominator scales the value of $\mathcal{L}_{\text{reg}}$ according to the amount of data points inside the search region. Using this relative quantity allows us to perform hyperparameter-free regularization over different search region sizes.
>
> In our experiments, we observed that adding $\mathcal{L}_{\text{reg}}$ did help avoid overfitting to large lengthscales and enabled better local estimation compared to SVGP (Figure 7 and 8 in appendix).

---

> ### Author Response · Authors · 2024-08-07
> **Rebuttal (2/4)**
>
> **Theoretical implications of sparse GP approximation**
>
> We agree that the theory section can be fleshed out more, and will do so by (1) providing more precise formulae for the additional regret incurred due to too-sparse approximations, and (2) experimental and numerical evidence for the increased approximation fidelity of Focalized GP.
>
> For the former, we do not aim to propose any new theorems due to the restrictive nature of common assumptions, lack of guarantees for ELBO-based methods, and difficulty for controlling the effective number of inducing points within the local region. However, we will better motivate our method by providing examples of high KL error under reasonable assumptions, e.g. those used in SVGP-TS[1].
>
> In **Figure R2**, we also empirically measure our claim that Focalized GP can significantly reduce approximation error on the search region. We sampled 8000 training points from 2d GP functions to train focalized GP and SVGP. Over different size of the search region, we compare the KL divergence of the GP posterior prediction over search region between sparse GPs and the exact GP. We observe that the KL divergence between focalized GP and exact GP is consistently smaller than that between SVGP and exact GP, implying tighter approximation to the exact GP over local region.
>
> While a rigorous regret bound is hard to derive, we conduct an empirical study where we directly compare the optimization performance between focalized GP and SVGP when combining with TuRBO. In this way we can eliminate the influence of hierarchical acquisition optimization. The optimization performances are shown in **Figure R3**. We observe that focalized GP outperforms SVGP on both high-dimensional problems, which empirically demonstrates our theoretical implications that Focalized GP contributes to reducing regret.
>
> [1] Vakili, Sattar, et al. "Scalable Thompson sampling using sparse Gaussian process models." Advances in neural information processing systems 34 (2021): 5631-5643.

---

> ### Author Response · Authors · 2024-08-07
> **Rebuttal (3/4)**
>
> **The novelty of our work**
>
> Here we highlight the novelty of FocalBO in both GP and BO aspects.
>
> **We design the first sparse GP model that improves acquisition function optimization**. Our design of focalized GP enables **better local estimation, aiming at improving acquisition function optimization during Bayesian optimization**. We adopt variational inference to derive focalized GP, which is **capable of performing joint posterior inference and sampling given a set of test points just like exact GP**. Therefore our proposed GP model is automatically compatible with any acquisition function that is used for exact GPs. (Figure 2).
>
> The locally induced GP (LIGP) in the cited work [2] is designed for regression tasks, which assigns different inducing points to every test point, aiming at improving the point estimation measured by MSE. The inability of joint posterior computation and discontinuous prediction prevents LIGP from easily being incorporated into the BO framework.
>
> **We propose the first scalable BO algorithm that is capable of utilizing large offline dataset for optimization.** Our design of hierarchical acquisition function optimization  with focalized GP **searches promising positions over different scale of search region during one BO iteration, enabling making decision based on both global and local information under restricted computational budget**.
>
> The overall idea of TuRBO [3] is to restrict the search space to a fixed size during one BO iteration, and adjust the search region size based on the optimization results. It achieves scalability by discarding previous samples and restarting when the search region reduce to certain threshold. Their use of exact GP still faces the scalability issue and would be computationally intensive when handling large offline datasets. As mentioned in the last paragraph of section 4.2, our proposed framework is orthogonal to TuRBO, and find that FocalBO and TuRBO can have a complementary effect in sections 5.2 and 5.3.
>
> [2] Cole, D. Austin, Ryan B. Christianson, and Robert B. Gramacy. "Locally induced Gaussian processes for large-scale simulation experiments." Statistics and Computing 31.3 (2021): 33.
>
> [3] Eriksson, David, et al. "Scalable global optimization via local Bayesian optimization." Advances in neural information processing systems 32 (2019).

---

> ### Author Response · Authors · 2024-08-07
> **Rebuttal (4/4)**
>
> **The rationality of FocalBO design**
>
> FocalBO has two main components: the focalized GP and the hierarchical acquisition optimization. In the clarification of the regularization term, we have explained that the scaling is used for maintaining similar scale for similar size of search regions, and our emprical evaluation demonstrate the rationality of our loss function design. Below we give some empirical results and the intuition in the design of hierarchical acquisition optimization.
>
> After each BO iteration, we use a simple yet effective heuristic to adjust the optimization depth for the next round, without introducing additional hyperparameters. We encourage more exploitation (increase the depth) when finding good points in the smallest search region, and encourage more exploration (decrease the depth) when good points are found in boarder search regions. On our set of diverse benchmarks - including synthetic and commonly-used test functions, as well as two challenging high-dimensional problems - this simple heuristic worked without any tuning required.
>
> **Comparison with original TuRBO**
>
> Our current experiments with TuRBO are all re-implementations to make sure baselines have the same computational cost. We run the original TuRBO implementation (with exact GP and Thompson sampling) on both robot morphology design and human musculoskeletal system control task (**Figure R4**). We observed that FocalBO outperforms TuRBO on both tasks with smaller computational cost. The reason of TuRBO's poor performance may be that it cannot quickly adapt over the search space when the online evaluation budget is small.
>
> **Typos and unclear statements**
>
> Thanks for pointing out the typos. We will correct them and improve our paper in the new version.

---

> > ### Comment · Reviewer_SUei · 2024-08-11
> >
> > I’ll reply to all 4 of your rebuttal messages here.
> >
> > Regarding Figure 1: Thanks for challenging me on this and I think you’re right.
> >
> > Regarding Lreg: Thanks for pointing out this alternative characterization of that cost function. I agree that heuristically this sort of makes sense as a way of dealing with some of the problems that arose. But what I’m suggesting is that it’s not convincing that this will work on general problems. The modification appears ad-hoc and appears not to be motivated by a statistical model, numerical approximation, or otherwise a precise idea of how big that term should be in order to balance against the expected log-likelihood and KL terms.
> >
> > Regarding the theoretical implications: Thanks for these simulations, they are interesting and help to build understanding of the proposed methodology.
> >
> > Regarding novelty of the focal approximation: I agree that there are fundamental differences between LIGP and your work, and I do not know of prior work which uses a “goal oriented” inducing point approximation for BO. I reference LIGP to point out that the idea of teleologically determining inducing points is not novel; it is rather its application to BO which is novel.
> >
> > Regarding Scalability: I must for now push back on the idea that the proposed method’s scalability is novel. The approach used in your article for scaling the GP falls into the meta-approach of approximating somehow the GP, and then doing the linear algebra exactly. It’s true that TuRBO does not use this meta-approach. However, an alternative meta-approach used for GPs is to avoid approximating at the statistical/GP level, but then to instead approximate the linear algebra directly. This latter approach is the one taken by Eriksson et al in their original implementation, which leverages pytorch and fancy linear algebra to avoid direct cholesky decomposition. Consequently, looking back at Eriksson et al, it seems like their largest dataset is of size 20,000. If I am not mistaken, the largest study in the proposed article in terms of total sample size is the Robot Morphology problem with a total problem size of 10,128, which is actually less. Furthermore, all but the last 128 of these are acquired offline. A priori, I would imagine that the fact that the ELBO has to be re-optimized at every level of depth makes for some amount of overhead. However, you did mention in your rebuttal that your algorithm had a smaller computational cost than TuRBO, though without providing the numerical results; I understand you may not have been able to fit those results onto the single rebuttal page. But on the basis of the observed experiments, we actually find that TuRBO’s original article contains the larger problems. (Please let me know if I missed or misinterpreted something).
> >
> > Regarding the comparison with the original TuRBO: Thanks very much for running these experiments, those are very helpful. It’s impressive that your method is able to keep ahead of it on those important applications.
> >
> > Regarding The rationality of FocalBO design: As I explained above, I still don’t understand why it would be the case that the Lreg term will always have the right magnitude relative to the Lwll and Lkl terms. But regarding the optimization depth criterion, I think I agree that what you’re doing makes sense.
> >
> > In conclusion: I thank the authors for the additional experiments and for engaging with my review. I think the direct comparisons against TuRBO are convincing so I’ll raise my score to a 4/borderline reject. Again, I think this is an interesting paper which clearly makes incremental progress on this challenging problem. Regretfully, my opinion is still that this paper does not have the novelty/impact necessary for a 2024 neurips publication (particularly because the large-offline setting, though certainly not contrived, I would argue is somewhat niche) but nor do I feel strongly that this article should be excluded from these proceedings.
> >
> > PS, upon reviewing the article, I noticed on line 188 that it’s not a stationary kernel that you’re looking for; stationary kernels which are periodic form counterexamples to your claim. Rather, it is kernels which decrease monotonically wrt some distance. (Just pointing it out for your benefit; no need for us to litigate this minor point and it does not impact my score; it's indeed clear that this projection will usually be efficiently computed in practice).

---

> > > ### Author Response · Authors · 2024-08-12
> > >
> > > Thank you for providing detailed feedback and raising the score!
> > > Below, we address your concerns regarding the scalability of FocalBO.
> > >
> > > As mentioned in our novelty rebuttal section, the scalability of TuRBO is primarily achieved through a restart mechanism, which discards all previously sampled data and re-initializes the optimization process. In our reproduction of the experiments from the original TuRBO paper, we found that the average number of evaluations in the rover planning problem (originally reported as 20,000 total evaluations) before restarting was $3198.7\pm649.8$ (mean$\pm$1 std, averaged over 77 restarts). This suggests that the actual data size used for GP in TuRBO is much smaller than data size in our addressed problem.
> > >
> > > Regarding the GP implementation in the original TuRBO, we agree that the conjugate gradient method and Lanczos decomposition in GPyTorch help to accelerate and parallelize computation via GPU.
> > > To further demonstrate the computational efficiency of our method, we ran FocalBO and the original TuRBO on an NVIDIA GeForce RTX 2080Ti with 11 GB of memory, instead of an NVIDIA A100 with 80 GB of memory. We observed that FocalBO was still capable of optimizing the robot morphology design task, while TuRBO encountered an out-of-memory error during GP training, even when using approximate computing methods. We believe that data size remains an issue in TuRBO, and this may be one of the reasons why TuRBO introduces a restart mechanism.

---

> > > > ### Author Response · Authors · 2024-08-14
> > > >
> > > > **Continue on scalibility**
> > > >
> > > > We argue that a direct comparison between the original TuRBO and FocalBO is not entirely fair. While focalized GP need to be optimized at every depth, it **share the same $\mathcal{O}(m^3)$ computational complexity as SVGP**. The $H\ll m$ re-optimization rounds do not alter this order. The computational complexity of TuRBO $\mathcal{O}(n^3)$ for using exact GP, and our empirical results demonstrate that using approximate linear algebra computations in GpyTorch does not reduce the computational cost to the same level as sparse GPs. We tracked GPU memory usage during the optimization of the robot morphology design task. The maximum GPU memory usage for the original TuRBO reached 78.9 GB, whereas FocalBO, with an inducing size of 200, used only 7.8 GB. Our emprirical evidence indicates that FocalBO is more scalable than original TuRBO.
> > > >
> > > >
> > > > **Impact of our work**
> > > >
> > > > We argue that our addressed problem setting is not niche. **The problems we tackle in the experiment section includes both large offline and large online setting, where FocalBO consistently demonstrates the superior performance.** In the human musculoskeletal system control task, FocalBO outperforms TuRBO with 3000 online evaluations, which we consider to be a substantial number.
> > > > We emphasize the importance of the large offline setting as no previous scalable BO methods have been able to effectively handle it. We believe our setting is important because the problems addressed by BO are typically expensive to evaluate. Instead of collecting data online from scratch, many problems already have data available from various sources. FocalBO’s ability to utilize larger datasets enables it to tackle challenging problems where the online evaluation budget is limited.

---

### Official Review · Reviewer_f3AD · 2024-07-12

**Soundness:** 4
**Presentation:** 3
**Contribution:** 3
**Rating:** 7
**Confidence:** 4

**Summary:**

This paper focuses on scalable Bayesian optimization, where the training data size is large and the search space is high dimensional. This work proposes a new kind of sparse GP model by designing a variational loss function that allows for adaptively focusing on interesting regions throughout the optimization process. By coupling this GP model with common acquisition functions such as EI, TS, PI, the authors show in experiments the effectiveness of this newly developed GP model. The authors also provide comprehensive algorithm analysis to further decipher the exploration vs. exploitation behavior of the algorithm.

**Strengths:**

The scalability issue of GP has greatly limited the application of BO to real-world problems. This paper introduces a novel sparse GP model and its complementary optimization strategy to solve this issue. The idea is intuitive and elegant, and with compelling empirical study, I think this is a valuable piece of work to the field.

**Weaknesses:**

Minor error:
- L130, "more that" -> "more than"

**Questions:**

- One concern is that the search region is always centered around the current best position. In the case when the best point from initially sampled points is far away from the true global optimum, and the landscape of objective function is not very smooth, then the algorithm will have a hard time to attain the global optimum quickly, because Focalized GP reduces to regular sparse GP in the broader search region and suffers inaccurate prediction. This issue might explain in Figure 3 why "FocalBO TuRBO" has a better performance than "FocalBO EI" (because TuRBO divided up the search space). I wonder if there is any better idea in setting the center of search region to encourage more exploration.

**Limitations:**

The discussion of limitations is absent in the paper, and it is better for the authors to discuss limitations of this work.

---

> ### Author Rebuttal · Authors · 2024-08-07
>
> Thank you for your appreciation of this work! We address your question below. The rebuttal figures can be found by opening the rebuttal PDF file.
>
> **Better way to center the search region**
>
> A plausible alternative would be to instead center the search region at the maximum of the chosen acquisition function. We empirically investigate this in **Figure R1 (a)**, which compares different ways of selecting the search region center by measuring the distance from the search region center to the global optima. We observe that current best point consistently is the closest to the global optimum, which validates this design choice.
>
> For the experiment above, we sampled 2d functions from GPs with Matern $\frac{5}{2}$ kernel and lengthscale of 0.05 (representing rigid functions), and selected the best point over unifromly sampled 10,000 points as the global optima.
>
> A sparse GP is already more explorative than using the full GP, since the smaller representational capacity leads to smoother posteriors. We demonstrate this empirically in **Figure R1 (b)**, where we measure the pair-wise distance of 100 Thompson sampling points under exact and SVGP (with 50 inducing points). We observe that sparse GP actually samples more diverse sets compared to exact GPs, i.e. exhibiting more exploration. Therefore, using focalized GPs does not sacrifice exploration, and significantly helps exploitation by performing acquisition function optimization over smaller search regions.
>
> Regarding Figure 3, we believe that the better performance of "FocalBO TuRBO" may be because utilizing trust regions helps FocalBO to better exploit at early optimization depths, leading to faster convergence with limited evaluation budgets.
>
> **Limitations of our work**
>
> While FocalBO demonstrate superior performances against existing scalable BO algorithms, theoretical can be further improved to make it more rigorous. For instance, an analysis of the convergence of FocalBO would enhance our understanding of the algorithm’s mechanism. The currently-used hypercube search region may not align well with the underlying function landscape, potentially wasting computational resources on low-performing regions. A more sophisticated search region design such as [1] may help FocalBO to further improve the optimization performance.
>
> [1] Wang, Linnan, Rodrigo Fonseca, and Yuandong Tian. "Learning search space partition for black-box optimization using monte carlo tree search." Advances in Neural Information Processing Systems 33 (2020): 19511-19522.

---

> > ### Comment · Reviewer_f3AD · 2024-08-13
> >
> > Thanks for providing empirical evidence for the design choice of search region center. I still think this work has adequate contribution to the field, however, regarding SUei's comment on novelty / impact of this work, the author did not offer more evidence, I would revise the score to 7 (Accept).

---

> > > ### Author Response · Authors · 2024-08-14
> > >
> > > Thank you for your feedback. Below, we clarify the novelty and impact of our work.
> > >
> > > **The novelty of our work**
> > >
> > > **We design the first sparse GP model that improves acquisition function optimization**. Our design of focalized GP enables **better local estimation, aiming at improving acquisition function optimization during Bayesian optimization**. We adopt variational inference to derive focalized GP, which is **capable of performing joint posterior inference and sampling given a set of test points just like exact GP**. Therefore our proposed GP model is automatically compatible with any acquisition function that is used for exact GPs. (Figure 2). Previous work which allocate inducing variables based on test points can not directly be incorporated into BO framework. Reviewer SUei also recognized that focalized GP is **fundamental different** from previous works.
> > >
> > >
> > >
> > > **We propose the first scalable BO algorithm that is capable of utilizing large offline dataset for optimization.** Our design of hierarchical acquisition function optimization  with focalized GP **searches promising positions over different scale of search region during one BO iteration, enabling making decision based on both global and local information under restricted computational budget**. Besides, FocalBO also demonstrates **superior optimization performance in large online setting** in our human musculoskeletal system control task.
> > >
> > > Regarding the Reviewer SUei's concerns of the scalibility, we emphasize that **our used focalized GP in FocalBO share the same $\mathcal{O}(m^3)$ computational complexity as SVGP**, while the complexity of TuRBO is $\mathcal{O}(n^3)$ for using exact GP. We addressed this point by investigating the restart mechanism of the original TuRBO and demonstrating a failure case of TuRBO under low computational resources.
> > > Our numerical experiment also demonstrates that TuRBO needs significantly more GPU memory than FocalBO even under approximated linear algebra computation (78.9 GB v.s. 7.8 GB). Our empirical evidences demonstrate that FocalBO is more scalable than orginal TuRBO.
> > >
> > > **The impact of our work**
> > >
> > > We argue that our addressed problem setting is not niche. **The problems we tackle in the experiment section includes both large offline and large online setting, where FocalBO consistently demonstrates the superior performance.** In the human musculoskeletal system control task, FocalBO outperforms TuRBO with 3000 online evaluations, which we consider to be a substantial number. We emphasize the importance of the large offline setting as no previous scalable BO methods have been able to effectively handle it. We believe our setting is important because the problems addressed by BO are typically expensive to evaluate. Instead of collecting data online from scratch, many problems already have data available from various sources. FocalBO’s ability to utilize larger datasets enables it to tackle challenging problems where the online evaluation budget is limited.

---

### Author Rebuttal · Authors · 2024-08-07

We would like to thank the reviewers for helpful and insightful reviews. In addition to addressing the comments below, we will incorporate the suggestions and new figures into the camera-ready version. Our additional experimental results are attached in the rebuttal PDF below.

---

### Decision · Program_Chairs · 2024-09-25

**Decision:**

Accept (poster)

**Comment:**

This paper proposes an approach for Bayesian Optimization in the large data regime that uses an SVGP with modified ELBO loss to improve quality of the surrogate in “promising” regions of the search space. Reviewers generally agree that the problem is relevant, and that the empirical performance of the approach compared to baselines is compelling (based also on the additional results provided by the authors during the rebuttal period). One of the main criticisms is that the methodology feels somewhat ad-hoc and that the theoretical analysis is underdeveloped. The former criticism is valid, but the approach is also simple, reasonably intuitive, and effective. There are some connections to other literature that I think are missing, in particular on the work of target-aware Bayesian inference (https://www.jmlr.org/papers/v21/19-102.html) whose goal is precisely to modify the inference procedure of the Bayesian model to improve its performance on the downstream task (in this case acquisition function optimization). I also should note that the following contemporary work on Approximation-Aware Bayesian Optimization using sparse GP approaches is very related https://arxiv.org/abs/2406.04308 (as this is only a recent preprint I am considering this as a miss of the authors, this is rather a heads up to help with improvements to the paper).

Overall, I think the quality of this paper is somewhat borderline - However, it’s one of the few that considers modifying the surrogate model for Bayesian Optimization in a way that targets the downstream task of better optimization performance, which in my opinion is under-studied and so this makes it a relevant contribution to the field. Hence I recommend it be accepted, but ask the authors to incorporate the constructive reviewer feedback and the additional empirical results.